# Nanotechnology Meets Oncology: Nanomaterials in Brain Cancer Research, Diagnosis and Therapy

**DOI:** 10.3390/ma12101588

**Published:** 2019-05-15

**Authors:** Alja Zottel, Alja Videtič Paska, Ivana Jovčevska

**Affiliations:** Medical Center for Molecular Biology, Institute of Biochemistry, Faculty of Medicine, University of Ljubljana, 1000 Ljubljana, Slovenia; alja.zottel@mf.uni-lj.si (A.Z.); alja.videtic@mf.uni-lj.si (A.V.P.)

**Keywords:** nanomaterials, nanoparticles, nanobodies, brain cancer, blood–brain barrier, glioblastoma, nanodiagnostics, nanotherapy, nanomedicine

## Abstract

Advances in technology of the past decades led to development of new nanometer scale diagnosis and treatment approaches in cancer medicine leading to establishment of nanooncology. Inorganic and organic nanomaterials have been shown to improve bioimaging techniques and targeted drug delivery systems. Their favorable physico-chemical characteristics, like small sizes, large surface area compared to volume, specific structural characteristics, and possibility to attach different molecules on their surface transform them into excellent transport vehicles able to cross cell and/or tissue barriers, including the blood–brain barrier. The latter is one of the greatest challenges in diagnosis and treatment of brain cancers. Application of nanomaterials can prolong the circulation time of the drugs and contrasting agents in the brain, posing an excellent opportunity for advancing the treatment of the most aggressive form of the brain cancer—glioblastomas. However, possible unwanted side-effects and toxicity issues must be considered before final clinical translation of nanoparticles.

## 1. Cancer: From Macro to Nano

Although advancement in medicine has been significant in the past decades, the early diagnosis, adequate therapy and favorable outcome of cancer treatment remain to be very challenging. Therefore, cancer still represents one of the major public health concerns, as it is complex, heterogeneous and a very aggressive disease [1,2]. According to the American Cancer Society, cancer accounts for about one in every six deaths globally. The number of new cancer patients in 2040 is expected to be as high as 27.5 million [1]. In other words, that would mean almost double from 14.1 million in 2012. The data for the US on a 5-year relative survival rate for all cancers combined show much better outcomes since the early 1960s. The percentage increased from 39% to 70% for Caucasian population, and from 27% to 63% for African Americans. Although the numbers seem very promising and they most likely reflect better understanding of tumor biology, advances in treatment, as well as earlier diagnosis for some cancers, a lot of cancer patients still do not reach these numbers [1,3,4].

In the past decade, several important developments in treatment approaches have been utilized in clinical oncology employing the patient’s genetic/genomic profile, immunotherapy and more targeted therapy [4]. With personalized medicine approaches, all these different aspects can be taken into account to tailor the treatment which would best suit the patient. In the course of technical advancement and development of medicine, macrosurgery has been replaced with microsurgery, and the latter accounted for lowering the mortality and morbidity rates [5]. Further advances in the field of medicine on the nanometer scale are about to offer even more precise surgery—the nanosurgery. Its application could be of great importance in removal of residual microtumors/single cancer cells that remain in organs after macrosurgery and represent a considerable threat for later lethal recurrence of cancer. In this context, several different nanoparticles are being tested for intraoperative detection and precise elimination of microscopic residual disease in vivo in real time [6,7]. 

The number of studies on nanomaterials is growing and it has been shown that their future in medicine is promising, as they have several exclusive physico-chemical and biological features like small sizes, large surface area compared to volume, specific structural characteristic, possibility to attach different molecules on their surface, capacity to cross cell and/or tissue barriers, and also long circulation time in the bloodstream. They can be used in several different biological applications from drug delivery systems, to contrasting agents, and diagnostic tools for highly specific detection of macromolecules [8]. In cancer treatment, it is of utmost significance to detect cancer as early as possible, since, at a very early stage, the cancer treatment is much easier. Namely, cancer cells in that stage have a decreased possibility to already possess mutations that account for development of drug resistance [6]. Nanomaterials represent an important breakthrough. Targeted cancer treatment in clinical trials using organic nanomaterials (liposomes and conjugates of different polymers and proteins) began in the mid-1980s while, in mid-1990s, they were initially released on the market [3]. Numerous nanomaterials are currently undergoing clinical trials; extended lists are available elsewhere [9,10,11,12,13].

Nanoparticles also present great potential for molecular cancer cell targeting and drug delivery. This will be exceptionally cherished and valuable in the oncology of the central nervous system (CNS), where the presence of the blood–brain barrier (BBB) is a big obstacle in the drug delivery process [5]. Treating CNS disorders is often ineffective because drugs cannot reach the target site [14]. The use of nanoparticles, however, may improve this problem as the small size of this particles enables them to cross the BBB and deliver drugs to the target site [14]. Neutral and anionic nanoparticles with sizes in the range of 20–70 nm have shown to cause less neurotoxicity and are preferential for transport across the BBB [10]. On the other hand, nanoparticles containing metals such as copper, silver and aluminium may result in greater neurotoxicity. In addition, nanoparticles are attractive for cancer imaging. For such purposes, nanoparticles with sizes less than 10 nm are highly desired due to their rapid renal clearance, which will result in smaller background signal [12].

In cancer treatment, nanobiotechnology is gaining an important role as it can be used for early detection, diagnosis and therapy. Further development of the field shows great potential also in *nanotheranostics*, which is integration of diagnosis and therapy through the use of the same nanoparticle, made of organic and inorganic materials, for both [2,6]. One of the challenges of nanobiotechnology is delivery of drugs to CNS, for which several different strategies have been developed and are extensively discussed elsewhere [5]. The focus of these strategies represents options for crossing the BBB as it opposes major challenges in drug delivery to the brain. Free drugs targeting the CNS are mostly administered systemically and their site of action may not be reached due to unsuccessful crossing of BBB, which results in insufficient drug concentrations in the brain in order to treat the disease. At the same time, techniques that circumvent the BBB and enable targeted delivery to CNS are invasive. They involve procedures like surgical implantation and infusion of devices, intraventricular or intracerebral injection of drugs [10]. In the brain, the distribution of the drugs is then dependent upon diffusion, which means that the concentration at the administration site is the highest and is decreasing further away from this site. However, this approach has the advantage of site-specific administration and thus minimizes the systemic toxicity. An interesting option for direct delivery to the brain is also transnasal administration through the olfactory bulb. On the other hand, the non-invasive administration methods are based on approaches that enable crossing the BBB, which have to consider the anatomical structure of the brain capillaries, transporter mechanisms, the extracellular matrix components, and also on the directed transport of the fluids across the brain [5,14,15,16,17].

### Glioblastoma

Malignancies of the central nervous system represent an enormous challenge for clinical oncology as well as for research. Patient mortality of the CNS cancers is one of the highest among all cancers as the CNS cancers have complex morphology and are relatively resistant to treatment [18]. In adult patients, gliomas represent the most common tumors. Histologically, they are classified as astrocytomas, oligodendrogliomas and oligoastrocytomas, while they are graded according to their degree of malignancy according to World Health Organization from Grade I to Grade IV. Among gliomas, glioblastomas represent the most common and lethal form of primary malignant brain tumor. It is the most aggressive form of glioma, which is commonly associated with discouraging prognosis for survival and fatal outcome within 12 months to 18 months after diagnosis. Currently, the combination therapies of surgery with temozolomide and radiation are used worldwide. In an effort to increase the positive patient outcome, alternative treatment options such as antibody–drug conjugate-based therapies and immunotherapy [19], as well as nanotechnology-based treatment options for glioblastoma [20,21], are extensively explored. There has not been a significant improvement of the medial survival of the patients, thus the urgent need for earlier diagnosis and more effective treatments [22,23]. In order to improve the sensibility of cancer detection and to achieve more effective treatment outcomes, the nanomaterials offer many advantages, also in the view of establishing personalized approaches in diagnosis and treatment of glioblastoma [4,24].

## 2. Nanomaterials in Biomedical Research

What are nanomaterials? Answering this question is not straightforward. In the European Union, the European Commission has recommended guidelines for nanomaterial definition, described in the 2011 Recommendations [25]. According to the guidelines, one of the external dimensions of 50% of particles should measure between 1 nm and 100 nm. In specific cases, 1%–50% of particles measuring between 1–100 nm could be accepted and also materials, such as fullerenes, graphene flakes and carbon nanotubes, that are smaller than 1 nm are considered as nanomaterials. However, determining dimensions in lower ranges remains an enormous challenge due to the limitations of microscopy, spectroscopy and single-particle inductively coupled plasma mass spectrometry [25]. Following guidelines of the US Food and Drug Administration (FDA), nanomaterials should meet two conditions: (a) size—at least one dimension should measure approximately 1–100 nm and (b) properties—their physical, chemical and/or biological features are a result of their size. Meeting the second condition, it is possible that the particles are larger than 100 nm [26]. In the period between the years 1970 and 2015, 359 nanomaterial drugs were submitted to the Centre for Drug Evaluation and Research, mostly liposomes, nanocrystals, emulsions and iron-polymer complexes.

So far, the most common application of nanomaterials is in cancer therapy [26]. Nanomaterials have a huge potential in research and medicine due to the unique characteristics as a result of their size. Their surface to mass ratio is large, they have longer half-life, reduce immunogenicity and can carry hydrophobic drugs. For every nanomaterial, there are three defining characteristics: its size, its shape and its surface [27]. Many different nanomaterials have been developed and extensively investigated such as metallic nanoparticles: silver, gold, iron and platinum, quantum dots, organic nanoparticles: liposomes, micelles, dendrimers and biological: nanobodies and extracellular vesicles (EV) [28,29,30,31,32]. The structures of different nanoparticles are schematically presented in Figure 1, while the properties of nanoparticles/nanomaterials are summarized in Table 1.

The following section will describe in more detail each of the nanoparticle/nanomaterial groups separately.

### 2.1. Metallic Nanoparticles

#### 2.1.1. Silver Nanoparticles

Silver has been historically known for its anti-microbial activity, therefore the first application of silver nanoparticles was in anti-microbial treatment [33]. Silver nanoparticles are formed by Ag(0) that is usually synthesized by reduction of Ag^+^. The methods of syntheses are chemical (reduction, electrochemistry, etc.), physical (vapour condensation, Arc discharge, etc.) and biological (during synthesis biological molecules, such as proteins and carbohydrates are used) [30]. To increase stability, the silver core is coated with polyvinylpyrrolidone (PVP) or citrate [33]. Kim et al. proved silver nanoparticle toxicity against *E. coli* and yeast, while its activity was mild in *S. aureus* [34]. The observed differences are probably owing to the different membrane structure as *S. aureu*s are Gram-positive and *E. coli* Gram-negative bacteria [34]. The mechanisms of silver function are not completely elucidated. It is known that silver nanoparticles damage the membrane [35]. They can enter bacteria and form complexes with sulphur- and phosphorus-containing molecules, e.g., DNA [36]. The damage is also caused by reactive oxygen species (ROS) formation [35]. Morones et al. investigated the mechanisms of anti-microbial activity in Gram-negative bacteria and showed that toxicity is dependent on size because only particles with size in the range 1–10 nm had an effect on bacteria [36]. Besides bactericidal function, silver nanoparticles can also kill viruses such as human immunodeficiency virus 1 (HIV-1), hepatitis B virus (HBV), respiratory syncytial virus (RSV) and the influenza virus. In addition, they were also investigated in cancer therapy [30]. Still, the major concern in silver nanoparticle use is toxicity that primarily depends on particle size, mode of synthesis and also coating [30].

#### 2.1.2. Gold Nanoparticles

Gold nanoparticles are produced by reduction of salts and are stabilized by phosphine, alkanethiol or citrate [37]. They are commonly surrounded by mixed monolayer protected clusters such as oligo (ethylene glycol) (OEG) or poly(ethylene glycol) (PEG) [37]. Gold nanoparticles have two unique properties—antibodies can be easily attached to surface, and plasmon resonance, the ability to absorb and scatter light of wavelength considerably larger than the particle [38]. One of the most studied mechanisms in drug delivery that exploits plasmon resonance is photothermal effect—particles are accumulated in tumors, radiated with the light of wavelength 800–1200 nm and locally release heat that destroys cancer cells nearby [37]. Cell destruction is usually caused by protein denaturation, nucleic acids breakage, membrane perforation and ROS generation [39]. It is especially important that gold nanoparticles absorb light in the near infrared region because the human body is transparent at those wavelengths [40]. Gold nanoparticles are used in both passive and active targeting. Passive targeting exploits the ability of gold nanoparticles to extravasate through the leaky gaps of blood vessels [37]. Impaired regulation of vascularization in tumors leads to the appearance of “enhanced permeability and retention” effect (EPR), which is aided by the increased pore size between endothelial cells (50–800 nm) in contrast to normal endothelium where pore size varies between 5–10 nm [41,42]. Active targeting includes binding of antibody and, when it reaches the selected location, it absorbs light during irradiation and induces heat. Gold nanoparticles can also carry drugs that are released when nanoparticles reach desired target and are irradiated. To evade the immune system, gold nanoparticles should be coated with thiolated PEG or liposome [38]. Furthermore, they are a suitable contrast agent in computer tomography (CT) imaging [43]. However, application of gold nanoparticles could induce unwanted toxicity which mainly depends on size and surface charge [40]. Particularly, particles with the size of 10 nm have been shown to accumulate in various organs (like blood, spleen and liver) and could be toxic to organs [40].

#### 2.1.3. Magnetic Nanoparticles

Magnetic nanoparticles are one of the most studied nanoparticles. Usually, Fe_3_O_4_ or γ-Fe_2_O_3_ form a core of particles that are additionally coated by polyvinylalcohol (PVA), dextran, PEG, polyvinylpyrrolidon (PVP) or chitosan [44]. They are usually produced by alkaline coprecipitation of iron (II) or iron (III) [45]. Both Fe_3_O_4_ and γ-Fe_2_O_3_ nanoparticles are supermagnetic, which means that they have magnetic properties only when an external magnetic field is applied [44]. When external magnetic field is applied, nanoparticles can travel to desired place [46]. In the blood, they are attacked by macrophages and monocytes. Their rapid clearance could be evaded by binding to PEG or other polymers [45]. When magnetic nanoparticles reach their target location, they are dispersed around tissue and, upon application of alternation of the magnetic field, release heat that destroys cancer cells nearby [47,48]. Cancer cells that usually reside in hypoxic regions are much more susceptible to elevated temperature than normal cells [47]. Usually temperature between 41 °C and 46 °C is generated. Nevertheless, to selectively target tumour without damaging healthy tissue remains a challenge [45]. Similar to other metallic nanoparticles, iron nanoparticle toxicity also raises concerns. The main risk poses free Fe^2+^ that, in reaction with hydrogen peroxide or oxygen, can form hydroxyl radicals and Fe^3+^ which in turn damage DNA and other molecules [44].

#### 2.1.4. Platinum Nanoparticles

Platinum nanoparticles are less studied as metallic nanoparticles, but have gained more attention as enzyme mimetics [32]. They could be used as a protection mechanism against ROS because they are very stable in the cellular environment and could serve as artificial catalase, glutathione peroxidase and superoxide dismutase [32].

### 2.2. Inorganic Nanoparticles

#### Quantum Dots

Quantum dots are 2–10 nm big semiconductors, usually inorganic nanocrystals composed of CdSe core and ZnS shell [49]. They are distinguished by their outstanding characteristics which are superior to the classical fluorescent dyes. Quantum dots are significantly brighter, more photostable, have broad excitation spectra and narrow excitation emission spectra [50]. This is especially useful in imaging since one source of light can be applied to excite many different colours [50]. Furthermore, quantum dots stay longer in the excited state and it is therefore easier to distinguish fluorescence of quantum dots from the background [51]. To solubilize, quantum dots can be bound to amphiphilic molecules or polysaccharides, be encapsulated in micelle or their surface ligands can be exchanged with thiol groups [51]. Due to the small size and EPR effect, they can penetrate tissues well, but need to be coated with PEG to evade immune system and prolong their half-life [50]. Quantum dots linked to tumor-specific antibody were applied in imaging of different cancers [51]. Their ability to absorb wavelengths of broad spectrum and emit narrow spectrum is especially beneficial, as one light source can be used and therefore the costs are significantly lower and the analysis of data is easier [52]. In addition, no signal amplification is necessary [52]. Moreover, it is possible to quantify signal, but comparison between individual signals is not possible [52]. Quantum dots can also be employed as a theranostic agent to carry drug and enable imaging simultaneously [53]. Nevertheless, application of quantum dots in medicine raises safety concerns. Under specific circumstances, cadmium in the core can release ions that are extremely toxic [50]. This can be avoided by suitable encapsulation or substitution by other elements, such as silicon, carbon or other cadmium free combination [50]. Silicon has been successfully applied in in vivo imaging and drug delivery as it is non-toxic and retains special optical characteristics at sizes less than 4 nm [54].

### 2.3. Organic Nanoparticles

#### 2.3.1. Liposomes

Liposomes, one of the most studied nanomaterials, are nano-scale spheres composed of either synthetic or natural phospholipid bilayers and aqueous cores [55]. Due to the amphiphilic nature of phospholipids, liposomes are formed spontaneously [55]. Liposomes are synthesized by thin-film hydration or reverse-phase evaporation [56]. They are either unilamellar (small—around 100 nm, large—200–800 nm) or multilamellar (500–5000 nm; consisting of many lipid bilayers having the same centre) [57]. Hydrophilic drugs are preferentially packed in unilamellar liposomes and hydrophobic drugs preferentially in multilamellar. Dynamics of release are different in each case [56]. Drugs such as doxorubicin are incorporated in liposomes by first exchange of buffers, acidic to neutral. Afterwards, the neutral drug can penetrate the liposome, is protonated due to an acidic environment and consequently cannot escape the interior of liposomes [58]. Other mechanisms of drug loading are liposome formation in the presence of saturated drugs and use of organic solvents [55]. Liposomes exploit the EPR effect [58]. The gaps in vessels allow up to 4000 kDa or 500 nm large vesicles to enter the tumor [56]. In tumor they can fuse with cells, are internalized by endocytosis or release drugs in extracellular space [56]. Liposomes can also be engineered in a way that release cargo upon suitable pH, redox potential, ultrasound and electromagnetic field [56]. They can target the tumor passively or actively (ligand mediated). Active targeting of tumors is not necessarily more efficient than passive targeting, but is advantageous in targeting vasculature, micrometastases and blood cancer [59]. Initial liposome application in blood confronted unwanted rapid clearance due to binding of serum proteins (opsonins) and consequent elimination by both mononuclear phagocyte system and complement system [60]. Liposomes were then further engineered and coated with PEG to increase biocompatibility, solubility in water, half-life of liposomes and to lower toxicity [60]. However, PEG coating does not completely diminish recognition by mononuclear phagocyte system [60]. To increase stability of liposomes, other options are available such as incorporation of cholesterol and phosphatidylcholine [56]. Liposomes have been extensively examined in therapy. It is known that size significantly affects half-life in blood and liposomes up to 100 nm penetrate tumors easier while liposomes of larger size have shorter half-life because they are better recognized by mononuclear phagocyte system [56]. To achieve active targeting, liposomes bind antibody-targeting tumor-specific antigen and deliver drugs to the tumor. Other options exist such as attachment of folate and transferrin because many cancer cells over-express folate and transferrin receptor [57]. Up to now, several liposomal drugs have been approved in medicine such as amphothericin B for systemic fungal infections, doxorubicin for metastatic ovarian cancer, AIDS-related Karposi’s sarcoma and metastatic breast cancer and vincristine for acute lymphoblastic leukemia [58,59]. Chemotherapeutics packed in liposome were also shown to evade multidrug resistance mechanisms [55]. Due to liposome superior characteristics, they are applied in other approaches such as gene delivery by packing DNA in liposomes that contain cationic lipids (e.g., DOPE or DOTMA) [60]. They are also an excellent imaging tool. In CT, they are applied to carry contrasting agent iodine and can detect both blood diseases and tumors [61]. Similarly, agent gadolinium can be packed in liposome to increase half-time and contrast at magnetic resonance imaging (MRI) [61]. Liposomes can also be formulated to contain air bubbles and be applied in ultrasound imaging [61].

#### 2.3.2. Block Copolymere Micelles

Block copolymere micelles are 10 nm–100 nm spheres formed by amphiphilic or oppositely charged copolymers and are classified into amphiphilic micelles, polyion complex micelles and micelles formed by complexing of metals [62]. Moderately hydrophobic micelles are formed by dissolving in water, while amphiphilic micelles are formed in organic solvent, e.g., dimethyl sulfoxide–DMSO, which must be removed afterwards [62]. Interactions leading micelle formation are hydrophobic interactions, electrostatic interactions and hydrogen bonds [63]. They consist of hydrophilic shell that controls pharmacokinetic properties and, inside, a hydrophobic core, that controls drug loading and release parameters [64]. The Hydrophobic core, that carries water-insoluble drugs, mostly consists of poly(propylene oxide), poly(esters), poly(L-amino acids) and phospholipids [65]. Shell usually consists of PEG, but other options have been considered, such as poly(N-vinyl-2-pyrrolidone) [65]. The main purpose to use micelle in drug delivery is to solubilize drugs that have (very) low solubility in water [65]. The advantages of micelle application are drug stability enhancement, protection against degradation and reduction of interaction with mononuclear phagocyte clearance system due to steric hindrance [65]. Micelles exploit the EPR effect and, compared to liposomes, are more stable and have higher loading capacity [64]. The main drawback in physiological conditions is dilution below critical micelle concentration, potential dissociation of micelles that consequently leads to potential severe side-effects [64]. Nevertheless, few drug-loaded micelles have entered clinical trials, such as paclitaxel, doxorubicin and cisplatin [64].

#### 2.3.3. Dendrimers

Dendrimers are globular synthetic macromolecules that consist of branches that stem from a core and form so-called generation (G) [66]. Their surface is polar and contains functional groups (positive, neutral and/or negative) [66]. Their unique characteristics are monodispersity and small size (5th generation (G5) dendrimer is approximately 5 nm big) [67]. Dendrimers are synthesized in divergent (building from the core to outside) or convergent (building from surface to core) manners [67]. Divergent mode is more common and has the advantage of enabling potential change of surface. On the other hand, high generation number dendrimers have potential defects [68]. Convergent mode is suitable when synthesizing smaller dendrimers [68]. Dendrimers usually consist of poly(amidoamine) (PAMAM), but their surface end is toxic and are therefore usually bound to PEG to decrease toxicity, enhance biocompatibility, increase half-life and enhance EPR effect [69]. In addition, two of the other most common dendrimers, poly(propylene imine) (PPI) and poly-L-lysine dendrimeres, show toxicity due to their cationic nature and, if they are negatively charged, the toxicity is lowered [67]. Moreover, dendrimers (G2-G4) are too small to exploit the EPR effect [68]. On the other hand, the advantages of dendrimers in drug delivery are slower release of drugs, lower toxicity and high accumulation in tumors [70]. In micelles, hydrophobic drugs can be packed into the interior of dendrimers while probes or other molecules can be bound to the outer surface of dendrimers [66]. For example, drugs such as paclitaxel and doxorubicin are bound to the core of dendrimers by hydrogen, hydrophobic and/or electrostatic interactions [71]. However, to achieve controlled release, drugs can also be covalently linked to surface and released into the environment upon selected stimulus such as pH, glutathione or specific enzymes [70]. Moreover, dendrimers can bind antibodies for active targeting [71]. Besides drugs, cationic dendrimers are suitable for nucleic acid delivery because they are hidden from enzyme degradation [71]. Dendrimers can also be applied in diagnostics and bind Gd as a contrast agent in magnetic resonance, iodine in CT or fluorescence probe and quantum dots in fluorescence imaging [66]. 

#### 2.3.4. Polymers

Polymer nanomaterials include polyplexes, polymer–drug conjugates and polymer micelles [72]. The advantages of polymers as drug delivery systems are improvement of pharmacokinetic and pharmacodinamic characteristics, increase of half-life and controlled release of drugs [73]. Similar to other nanoparticles, polymers also exploit the EPR effect [73]. In order to carry drugs, polymers should be non-toxic, non-immunogenic and have a suitable loading capacity [73]. The most common carriers are N-(2-hydroxypropyl) methacrylamide (HPMA) PEG, poly(glutamic acid), polyethyleneimine (PEI), dextran, chitosan, poly(L-lysine) and poly(aspartamides) [73]. The natural polymers suitable in nanotechnology applications are chitosan, sodium alginate, gelatine and starch [74]. Polymers can release drugs upon diffusion-control, solvent-activation, chemical-control or external triggers (temperature, pH, redox potential) [73]. The enormous attention as polymer carrier has gained poly lactic-co-glycolic acid (PLGA), a copolymer of poly lactic acid and poly glycolic acid, because it is biodegradable and biocompatible [75]. The main biodegradation process of PLGA is hydrolysis that is mainly dependent on molar ratio between lactic and glycolic acids, molecular weight of polymer and crystallinity [75]. At the first phase of release, drugs on surface are released due to solubility and water penetration into polymer. In the second phase, the water hydrolyses the polymer and consequently the drug is released from the depth of the polymer [75]. The second of the most studied polymers in nanomedicine application is polybutylcyanoacrylate (PBCA), favoured due to non-toxicity and degradability [76]. In the body, it is degraded by esterases [76]. Drugs, carried by BCA, can be released by several mechanisms such as desorption of the surface, diffusion from matrix or wall of nanocapsules, degradation of matrix or combination of diffusion and degradation [76]. Polymer surfaces could be additionally modified to avoid mononuclear phagocyte system, prolong half-life, enhance bioavailability or to achieve active targeting [76]. The modification with Polysorbate 80 (Tween 80) has been extensively studied in brain diseases because it enables penetration through the blood–brain barrier (BBB) [76]. The potential mechanisms of penetration are (a) inhibition of p-glycoprotein; (b) degradation of polymer and consequent damage of BBB and permeability enhancement; and (c) endocytosis by brain vessel cells [76]. Besides drugs, polymers can serve as gene carriers, especially PEI, poly(L-lysine) (PLL) or degradable polycations [72].

### 2.4. Biological Nanoparticles

#### 2.4.1. Nanobodies

Nanobodies are 15 kDa large, 4 nm long and 2.5 nm wide antigen recognizing parts of heavy-chain only antibodies produced by *Camelidae* [77,78]. Their scaffolds are formed by two β-sheeted structures and contain four framework regions and three complementarity-determining regions (CDR) [77,79]. The principal protocol of nanobody generation is first immunisation of llama/alpaca, obtaining lymphocytes, generation of phage display library of nanobodies and panning to obtain nanobodies against desired targets [80]. Affinity and specificity of nanobodies from immune libraries are very good as the dissociation constant can even be in the nano- or picomolar range [81]. They have many advantages over classical antibodies. Their production is easy as nanobodies can be produced in microbial systems such as *E. coli* or yeasts with high yield [77]. Due to their small size and long CDR3, they can bind to epitopes that are hidden to classical antibodies [77]. They are also highly stable at elevated temperatures (their melting point is 67–78 °C) and non-physiological pH [77,82]. Moreover, particular hydrophobic amino acids in the framework-2 region are substituted by hydrophilic amino acids which renders them water-soluble [77]. Due to their structural similarity to human immunoglobulins, nanobodies present with low immunogenic potential. In addition, additional non-immunogenicity can be achieved by humanisation [82]. However, in clinical trials, cases of developing anti-nanobody antibodies have been reported, but those nanobodies did not have impact on safety [79]. In therapy, the size of nanobody is disadvantageous because it is below renal clearance limit. Therefore, its half-life should be increased, for example by binding to other nanobodies, albumin or PEG [82]. On the other hand, rapid clearance my result in lower toxicity [83]. In vivo, nanobodies are able to reach tumors from vessels by diffusion and help of fluid transport. In comparison to classical antibodies, they penetrate tumors better and their tumor distribution is more homogenous [83]. Only a few nanobodies in cancer treatment have entered clinical trials such as bivalent nanobody ALX-0651 against C-X-C chemokine receptor 4 (CXCR4) and tetrameric nanobody TAS266 against death receptor 5, but both of the clinical trials have been discontinued due to low efficiency or toxicity [79]. However, nanobodies against a large spectrum of diseases like rheumatoid arthritis, systemic lupus erythematosus, psoriasis and breast cancer are currently undergoing clinical trials (please visit https://clinicaltrials.gov and search keywords “nanobody”, “nanobodies”, “VHH” and “Ablynx”). In diagnostics, nanobodies have been successfully bound to fluorophore such as IRDye800CW and directed toward a specific target. The method benefitted from high tumor to background ratio [84]. Their short half-life is also suitable in positron emission tomography (PET) and single photon emission computed tomography (SPECT) and promising results have been observed by nanobodies bound to 18 F or 68 Ga [84].

#### 2.4.2. Extracellular Vesicles

Extracellular vesicles (EV) were first known as cell waste. Now, they present one of the most important means of communication between cells [85]. Currently, several different extracellular vesicles are known such as exosomes, microvesicles, apoptotic bodies and oncosomes [86]. Exosomes are the smallest EV. They measure 40 nm–120 nm and are formed by fusion of multivesicular bodies to membrane [86]. Microvesicles are, on the other hand, larger (50–1500 nm) and are formed by budding from plasma membrane [87]. EV carry several biological molecules such as proteins, nucleic acids and lipids and can travel to different organs by body fluids [88]. Regarding RNA, mostly miRNA and tRNA are found in EV, but their composition does not necessarily reflect cell profile [86]. Protein cargo in EV is extremely heterogeneous. Exosomes contain proteins that are characteristic to endosomal compartments, such as MHC II, CD37 and CD63. In addition, they contain proteins that have functional roles [86]. When EVs reach their target cell, they can be internalized by fusion or endocytosis. Several molecules contribute to better internalizations such as interaction between tetraspannin proteins on exosomes and integrins on target cells, phosphatidylserine on exosomes or heparan sulphate proteoglycans on target cells [88].

## 3. Nanomaterials and Brain Cancer 

As the brain is one of the most important, complex and specialized organs, it has to be extraordinarily protected. It is secured by two main barriers: the blood–cerebrospinal fluid barrier and the blood–brain barrier [16]. The BBB serves for dynamic transport of nutrients, peptides, proteins and immune cells between the brain and blood [15]. The BBB is formed by the tight junctions between the microvascular endothelial cells of the CNS and it works through a selective permeability system. Paracellular transport between the endothelial cells and transcellular transport (active and passive) through the luminal side of the endothelial cells, across the cytoplasm to the abluminal site of the endothelial cells, and finally to the interstitium of the brain, represent the main pathways for crossing the BBB of small molecules. In order to treat brain disease, the therapeutic has to be able to cross the BBB and at the same time it must not cause immune response [16,89]. The mechanism of a drug delivery across the BBB should ideally be well controlled, selective and not cause any damage to the BBB. The carrier should target the BBB, although the administration is systemic; at the same time, the carrier should be biodegradable and not toxic. The concentration of the drug should reach the therapeutic concentration at the site of action and remain for an appropriate time span to achieve the desired effect [15].

Use of nanoparticles for delivery to the brain represents a promising approach, since they are small and could therefore cross the BBB and deliver drugs to their action site [5,14]. In association with potential drug delivery to the brain, several different nanomaterials have been studied, including, but not limited to, gold nanoparticles, liposomes, dendrimers, carbon nanotubes, and micelles (Figure 2) [16].

Delivery of the drugs through the BBB is associated with the ability of nanoparticles to mask the BBB-limiting characteristics of the drugs, meaning that dugs are delivered to the brain through the BBB transport mechanisms. This is of great importance, since, with targeted delivery systems, chemotherapeutic drugs like paclitaxel can be delivered to the brain in concentrations lower than standard therapeutic doses of the free drug, which results in safer drug dosing while still achieving desired therapeutic efficacy [14]. Important advantages of the nanoparticles in treatment and diagnostics are also biodegradability, controlled loading and release of the transported drugs, stability of the nanoparticles and shelf life, and the non-invasive approach [17]. For transport of many molecules, inhibition of transmembrane efflux through systems like P-glycoprotein can also be overcome with appropriate assembly (coating) of nanoparticles [15]. While organic particles are mostly used for drug delivery and treatment, the inorganic nanoparticles have been mostly used for different diagnostic bioimaging techniques for improved tumor visualization by enhancing the contrast between tumor and healthy tissue and in some cases also as effective drug delivery systems [16].

### 3.1. Mechanisms of Nanomaterials Transport across the Blood–Brain Barrier

Although the main role of BBB is regulation of homeostasis and protection of the brain from harmful blood-borne substances, microorganisms and hormones, for therapy, it is an obstacle as it prevents almost all high molecular weight drugs and 98% of the low molecular weight molecules to cross it [90]. There are different methods for increasing transport across the BBB like disruption with osmotic shock, ultrasound and magnetic gradient [91]. These physical methods that damage the integrity of the BBB and increase its permeability are invasive and can cause irreversible CNS damage. Another method is the use of cell penetrating peptides (CPPs). CPPs are less than 30 amino acids long biologically safe peptides with low toxicity. They can be used as effective carriers for drug delivery and can deliver small as well as large macromolecules including proteins, nanoparticles and nucleic acids [92]. Modifications of nanoparticles with CPPs show increased brain delivery efficiency, promote translocation of cargo into cells and increase particle uptake [91]. In general, increasing blood circulation time usually helps in delivery of nanoparticles to the brain. Other possible mechanisms are adsorptive and receptor-mediated transcytosis, clathrin-mediated and caveolin-mediated endocytosis [93,94]. However, receptor-mediated endocytosis/transcytosis is the most effective transport mechanism for endogenous macromolecules. It takes place on the blood side where drugs bind to specific receptors and form corpuscules through endocytosis; then, the drug moves across the cytoplasm of the endothelial cell and, from there, it is exocytosed in the brain capillary endothelium where it plays a biological role [90]. As the transferrin receptor (TfR) is highly expressed in the brain capillaries, natural ligand transferrin (TF) can be used as a carrier for drug delivery. One study showed that loparemid-loaded human serum albumin nanoparticles with covalently bound TF are able to transfer the loparemid across the BBB [95]. Other receptors which can be used for receptor-mediated transport are insulin receptor expressed on the surface of capillary endothelial cells in the brain and low density lipoprotein receptor-related protein 1 and 2 ligands which are highly expressed in glioma cells [91]. Nanoparticles that cross the BBB are mostly polymeric and include PLBA, PLGA and poly(lactic acid) (PLA), but also liposomes, gold, silver and zinc oxide nanoparticles [96]. In in vivo study using mouse models, Betzer et al. show that nanoparticles with sizes between 20 nm and 70 nm enable visualization and localization of the particles with CT [97]. The authors also show that insulin-coated nanoparticles accumulate in the brain in higher amounts than the other investigated nanoparticle types (5% compared to 0.5%, respectively). In another in vivo study using glioblastoma-bearing rats as a model, PS80-coated nanoparticles loaded with doxorubicin showed significant antitumor effect, which suggests they are able to cross the BBB and release the cargo in the brain parenchyma [98]. In a different study, anti-EGFRvIII antibody conjugated to iron nanoparticles extended the survival of glioma-bearing mice [99]. Overall, when designing nanoparticles, it should be considered that positively charged nanomaterials with sizes smaller than 200 nm and decorated surface with different ligands are more likely to successfully cross the BBB [93]. 

### 3.2. Use of Nanoparticles in Glioblastoma Targeting

So far, nanomaterials have been used for detection, diagnosis and treatment of brain tumors. Current diagnosis of glioblastoma is based on brain imaging, with resolution of 1 mm. With the introduction of nanomaterials, the resolution and consequently earlier diagnosis could be improved by the factor ten or even more. However, the nanoparticles for this purpose still have to be improved in their accumulation and retention times within tumors [24]. In bioimaging with magnetic resonance different forms of superparamagnetic iron oxide nanoparticles have been investigated as contrasting agents. It has been shown that, when superparamagnetic iron oxide nanoparticles are coated with bovine serum albumin and further conjugated with folic acid, which is a tumor-specific ligand, it can be used for tumor-targeted contrasting agents. Further labelling with fluorescein isothiocyanate enabled also intracellular visualization [17,100]. Magnetic characteristics of inorganic nanoparticles can also be used in treatment through the induction of hyperthermia. Magnetic hyperthermia is based on a single intratumoral injection of a large portion of magnetic nanoparticles. These produce heat after being exposed to an external alternating magnetic field, which results in cell death induction. In clinical treatment of different cancers, mostly those representing terminal illness and also in unresectable tumors, which can both be the case in glioblastoma, magnetic hyperthermia has already proven to be an effective approach. A very promising characteristic includes iron oxide nanomaterials, as they can be used in multiple ways at once—like therapy and diagnosis and also in multiple therapeutic strategies, like magnetic and photothermal strategies. Besides the direct effect of hyperthermia through inducing cell death, it can be also linked to restoration of cancer-specific immune response [101,102]. Like iron oxide nanoparticles, gold nanoparticles can also be used in multiple ways. Besides being an effective drug transport system for treating glioblastoma as they can carry chemotherapeutic doxorubicin [16], it has been shown that, after modification with transactivator of transcription (TAT) peptide, they can deliver both therapeutic (doxorubicin) and diagnostic (gadolinium chelates that are the most widely used contrast in clinical setting) agents across the BBB and into the tumor [103]. 

On the contrary to inorganic particles used mostly for imaging and hyperthermia treatment approach, and less for drug delivery, the latter is mostly based on organic nanomaterials like liposomes, micelles, dendrimers and polymers. Liposomes represent a promising drug carrier, as they are nontoxic and nonimmunogenic; however, their transport through the BBB remains challenging because some of the drugs may also be released within the BBB. Encapsulating nerve growth factor into the liposomes promoted their transport across the BBB. Further in vivo experiments on animals showed that modification of liposomes with short cell-penetrating peptides of eight arginines (R8) conjugated with oleic acid leads to assembly of effective carriers for doxorubicin, which currently represents an alternative treatment option to temozolomide. The reuptake of modified liposomes in the brain was 2.4 times higher than that of the liposomes without modifications; however, the half-life of both types of the liposomes remained similar [104]. Use of immunoliposomes, which are antibody-directed liposomes, has been shown to significantly increase the transport of daunomycin into the brain mediated with the monoclonal antibody for the epidermal growth factor, which has been shown to be overexpressed in glioblastoma [105].

An interesting approach for improving the accumulation of cancer drugs in the cells is through the use of ligands that facilitate the translocation on nanoparticles across the endothelium of the tumor vessels, such as cyclic peptide composed of Arg-Gly-Asp (cRGD) with selective affinity for the α_v_β_3_ and α_v_β_5_ integrins, which show overexpression on the endothelial cells of tumor angiogenic vessels. In vivo experiments showed efficient drug delivery through the use of cRGD ligated to polymeric micelle bearing platinum anticancer drug. The nanoparticle had high permeability from vessels into the tumor and has significant antitumor effect in glioblastoma [106].

For drug transport through the BBB, polymeric nanoparticles can be used. PBCA coated with polysorbate 80 can carry different molecules like peptides, hydrophilic and lipophilic compounds. In the blood, these nanoparticles are absorbed on the apolipoproteins. In the BBB, they interact with lipoprotein receptors on the endothelial cells of the brain capillaries. [14]. They can be loaded with cytostatics like doxorubicin and, after the transport across the BBB, they can reach therapeutic concentrations in the brain. In animal studies, they showed 20–40% of tumor remission and more importantly reduced the dose-limiting cardiotoxicity [24,107]. Another polymer with certain biodegradation manner and improved mechanical properties, PLGA has been loaded with irinotecan hydrochloride (IRI), a topoisomerase I inhibitor and potential chemotherapeutic or metformin hydrochloride (MET). IRI inhibits DNA topoisomerase I and causes DNA breaks that result in apoptosis, while MET activates AMP-activated protein kinase (AMPK) and therefore inhibits the mechanistic target of rapamycin (mTOR) pathway. For both IRI and MET-loaded nanoparticles, a significant reduction in tumor volume was present in comparison to the use of free drugs. The use of these nanoparticles poses a promising approach towards glioblastoma treatment [108]. An alternative to synthetic polymers is also lipid-based polymers that are made of highly biocompatible and biodegradable natural components. These very stable carriers loaded with temozolomide confirmed cytotoxicity and inhibition of tumor growth in vitro and in vivo experiments, respectively, while showing a sustained drug release profile [109]. The same group of researchers showed also a good potential of these nanoparticles for dual-targeting of glioma with temozolomide and vincristine. Again, in vitro and in vivo studies resulted in tumor cells inhibition and tumor regression, suggesting these nanoparticles as an excellent delivery system for glioblastoma treatment [110]. Further modification of nanostructured lipid carriers with RGD peptide for targeted delivery and temozolomide therapy leads to efficient and selective delivery of the drug to glioma cells, followed by effective inhibition of tumors [111]. 

Dendrimers have so far been utilized for targeting glioblastomas. Namely, dendrimers have a well-defined structure and are suitable for transporting more than one therapeutic through the BBB and to the brain. For in vitro experiments, a more complex pH-sensitive dual-targeting 4th generation PAMAM dendrimer G4-DOX-PEG-Tf-tamoxifen has been constructed. On the inside, it was bearing tamoxifen and on the outer surface was conjugated transferrin (Tf) enabling receptor-mediated transport across the BBB. In the experiment, it was shown that this nanoparticle could cross the BBB and induce the inhibition and death of glioma cells [112]. Application of in vivo arginine-modified polyamidoamine dendrimer (PAMAM-R), which was delivering a therapeutic gene, human interferon beta (IFN-β), was injected into xenograft brain tumors of animals. The effect was significantly smaller tumor size than in control animals, further confirmed with immunohistochemistry that showed inhibition of tumor growth. Finally, the RT-qPCR confirmed the expression of IFN-β and therefore induction of apoptosis in vivo. Based on the selective apoptotic effects on tumor cells, this approach represents a potential in glioblastoma treatment [113].

EV in brains have several roles in both physiological and pathological states. In Alzheimer’s disease, EV released by neurons stimulate degradation of beta-amyloid [114]. Exosomes also play a role in viral infections. For example, HIV infected astrocytes release exosomes that contain viral proteins, such as Nef [115]. EV could be employed as a natural nanoparticle system in brain cancer treatment. They are biocompatible, less toxic and immunogenic. Moreover, in comparison to liposomes of the same size, they can target cells 10 times more efficiently [116]. In therapy or for other purposes, many different molecules can be packed in exosomes such as RNA and chemotherapeutics [117,118]. In brain cancer therapy, first they need to pass the blood–brain barrier. Several mechanisms of penetration have been described already. Exosomes derived from brain microvascular endothelial cells can penetrate the impaired blood–brain barrier, but not the healthy one. Exosome can also be linked to specific molecules that enable transport through BBB, such as rabbies virus glycoprotein [114]. Exosomes have already been tested in vivo to target glioblastoma. They were derived from embryonic stem cells and have been modified with c(RGDyK) peptide and packed with paclitaxel. Results were very promising as mice with brain tumor xenograft targeted with c(RGDyK) and paclitaxel had considerably higher mean survival in comparison to control [119]. Exosomes also present a huge potential in the diagnosis of glioma because they are present in blood and cerebrospinal fluid. They contain enriched tumor-specific proteins. Up to now, several exosomal potential biomarkers have been proposed such as RNA encoding EGFRvIII and IDH1 and miR-21 [120].

In glioblastoma, theranostic approaches are being developed. Two recent studies on liposomes showed potential for improved in vivo imaging and inhibition of tumors. Xu et al. assembled a long-circulating liposome integrating SPIONs (superparamgnetic iron oxide nanoparticles), quantum dots and integrin antagonist cilengitide [121]. These nanoparticles enable obvious negative-contrast enhancement effect in dual-imaging (magnetic resonance—near infrared imaging) studies and at the same time treatment of glioma. Another complex liposome-based nanoparticle for multimodal imaging (CT/MRI/fluorescence) with photothermal therapy approach was assembled by Wu et al. [122]. It showed complete inhibition of tumor growth and represents a promising platform in nanotheranostic approaches. Additionally, gadolinium nanoparticles are being used as theranostics, which means they are at the same time serving therapeutic applications and diagnostic tests in glioblastoma [16].

Currently, for glioblastoma, there are 10 nanomaterials which are being investigated for possible clinical use as presented in Table 2. Data were obtained through searching the clinical trial registry https://clinicaltrials.gov and keywords “micelle”, “nanoparticle”, “liposome”, “nanomedicine(s)” (access date 23 April 2019).

In general, clinical trials are divided into the following phases [12,123]:Phase I: dosing, toxicity and excretion in healthy subjects;Phase II: safety, efficacy in large patient groups;Phase III: multi-centered randomized placebo-controlled trials;Phase IV: post-marketing studies, requested by health care professionals or the FDA.

Although early diagnosis of cancer is favorable in many aspects, there are some cases when detection of tumors does not mean the presence of an advanced disease, namely, many neuroblastomas can exist in patients for a long time as indolent tumors. Since treatment of these cases is not fully understood, it has to be discussed very thoroughly in which cases these tumors should be treated and in which cases they should be left alone [124]. Additionally, when considering the results of different studies, the results have to be considered with caution. Namely, many studies are performed on the subjects with different diseases, including brain tumors, which all can affect the BBB. Many of the studies on nanomaterials as plausible carriers for detection, diagnosis and treatment of brain tumor are performed on animals, and therefore the potential to translate the approaches on humans is limited [15].

## 4. Nanotechnology

Nanotechnology is an elegant term for conducting research and technology development at atomic, molecular and macromolecular levels; intentional design, construction and utilization of materials, devices, structures and systems through controlling and manipulating molecules that have at least one characteristic dimension measured in the nanometer range (1–100 nm); and creation and use of structures and devices that have novel functions as a result of their small(er) size and are able to control at an atomic level [125,126,127,128]. In general, nanoparticles are considered molecules with sizes in the range from 1 nm to 1 µm. However, for biomedical purposes, they are usually in the range between 10 nm and 800 nm. The implementation of nanotechnology in life sciences will result in the formation of nanobiotechnology [129]. Nanotechnology shows strong potential to influence the field of biomedicine, with emphasis on enabling early disease diagnosis and improving treatment methods [126]. The greatest impact of nanotechnology will be in the management of cancer, neurological disorders and cardiovascular diseases [6]. The physical properties of nanoparticles (size, shape, charge and surface) are crucial for designing successful therapeutic strategies.

### 4.1. Nanodiagnostics

In general, nanodiagnostics is used to describe the use of nanotechnology for molecular diagnostic purposes. Nanodiagnostics refers to the planned construction of devices using at least one dimension on the nanoscale for detection of events that spontaneously happen on the nanoscale [125]. Nanodiagnostics evolves to meet the demands of clinical diagnostics for early disease detection and high sensitivity [126,130]. Where applicable, nanodevices can be used in combination with existing imaging technologies for more precise in vivo and in vitro diagnosis [130]. The use of nanotechnology for molecular diagnostics is usually broadly categorized as “biochips/microarrays” or more precisely “nanochips/nanoarrays” [125]. Nanodiagnostics will enable detection of biological molecules which are below the detection limits of conventional techniques and will give information about individual members in the heterogeneous biological systems and their lifespans [5]. Nanodiagnostic-based platforms are currently developed for the detection of biomarkers for genetic diseases, single nucleotide polymorphism (SNPs) and pathogen nucleic acids [126,130].

So far, various nanoparticles have been developed for use in diagnostics with the most common being gold nanoparticles, dendrimers and quantum dots [101,129]. For example, small DNA fragments can be attached on gold nanoparticles which are then used for simultaneous detection of different DNA fragments [5]. In this case, assembly on the sensor surface happens only in the presence of complementary DNA. It is expected that these tests will have higher sensitivity than conventional enzyme-linked immunosorbent assay (ELISA) [130]. In addition, DNA-labeled magnetic nanobeads have the potential to detect any nucleic acid or protein with a lot higher sensitivity than current technologies allow for. Dendrimers, synthetic polymers with branching tree-like structures, are another type of nanodevices which can detect multiple proteins or peptides which are valuable indicators of different physiological phases [101,130]. Dendrimers can be used as a platform for targeted delivery of chemotherapeutics in cancers [131]. They can be used for predicting treatment efficiency by monitoring cell apoptosis caused by the chemotherapy [131]. Dendrimers usually contain metal nanoparticles which are toxic for humans. Combination of dendrimers with gold nanoparticles are more suited for use in living organisms. Additionally, biocompatible dendrimers are being developed and surface modification with PEGylation, acetylation and glycosylation are proposed to reduce toxicity [131]. Moreover, the inorganic fluorophores quantum dots are the most promising nanostructures for diagnosis. Quantum dots can also be used for performing sentinel lymph node biopsy—identification of cancer cells in a single lymph node [131]. This allows surgeons to identify the diseased lymph node without the necessity to remove the patient’s entire lymph system. Biomolecule-coated ultra-small super paramagnetic iron oxide particles and magnetic iron-oxide nanoparticles have been used as contrasting agents for MR cancer detection and imaging, respectively [128,130,131]. Contrary to gadolinium-based agents which rapidly diffuse due to the leaky tumor vasculature and give blurry images, iron nanoparticles localize within the neoplasm and give images with sharp margins [128]. They can be attached to cells without affecting cell proliferation, differentiation and function [131]. Such technologies will enable cancer detection earlier than the conventional techniques allow for it. They can also be used for detection of metastatic and residual disease. The ultra-small multimodal nanoparticles named cornel dots are developed to show the extent of tumors in human organs [131]. They can reveal the magnitude of tumor blood vessels, show invasive/metastatic spread to distant organs and track cell death and treatment response [131].

Nanodiagnostics have a long way to go before entering into the routine clinical practice. Translation of the research results into health care services is a complex process which involves rigorous performance evaluation, as well as validation, screening and testing of the basic research results so they can prove valuable and reliable to be used in the clinic (i.e., to avoid the detection of false positives). The use of nanotechnology for diagnostics is further complicated by the current lack of knowledge about the interactions of nanoscale materials with biological systems. Once mature enough to enter the clinic, nanodiagnostics will revolutionize our view of molecular diagnostics due to the improved sensitivity without loss of specificity, use of small sample amounts, which is critical for human samples, and reduced amounts of reagents, which will result in lower costs. Another benefit is the use of nanodiagnostics as a non-invasive method for diagnostic purposes and disease monitoring, especially at the single cell level. Single cell analysis referred to as “single cell genomics” allows for analysis at a cellular level without the need for amplification or cloning, which leads to faster results—a characteristic that is important for diagnostic purposes [6]. This will enable disease detection in early developmental stages that are still manageable. It is reasoned that increase in patient survival can be reached with sensitive diagnostic systems that can be used as early warnings even before appearance of clinical symptoms [124].

### 4.2. Nanotherapy

Existing cancer therapy consists of surgery, radiation, chemotherapy and immunotherapy [41]. Current therapies are not always successful in treating diseases due to the lack of effective drug delivery to the desired site [5,132]. Failure is also a result of pharmacologic or toxicity issues and acquired drug resistance [133]. One of the major issues of conventional chemotherapeutic agents in this regard is drug solubility, which is often poor in aqueous solutions. This is why they require to be dissolved in toxic solubilizing agents for parenteral administration [123]. Correspondingly, in order to limit systemic toxicity, the dosage has to be lowered. As a result of the advantageous properties of nanoparticles and nanomedicines such as better water solubility, smaller size, improved stability and shelf life, increased bioavailability and release rates as well as targeted delivery of biological therapies, nanotechnology can significantly improve drug delivery and consequently therapy. Nanoparticles or nanocarriers (nanomaterials that are used for the delivery of anticancer drugs) can boost the current therapeutic windows and clinical success in disease management with their superior properties: better pharmacogenetic features, longer blood circulation time, cellular uptake and distribution volume [41,101].

Biological therapies include the use of vaccines, cell therapy, gene therapy, antisense therapy and RNA interference as therapeutics [5]. Briefly, DNA vaccines can be used against pathogens like human immunodeficiency virus (HIV), hepatitis C, tuberculosis and malaria—nanoparticles can optimize their delivery systems; cell therapies can be used for prevention of human diseases by administration multiplied and pharmacologically treated/altered cells outside of the human body—nanoparticles can improve labeling, visualization and monitoring of such cells; and in gene therapy, which is the transfer of genetic material to specific cell targets, nanoparticles can be used for non-viral gene delivery. In general, targeted nanoparticle-based drug delivery will increase drug accumulation within the tumor which will reduce the required dosage amount and consequently decrease unwanted side-effects [123]. One of the methods for targeting solid tumors is by aiming at their microenvironment. Tumor microenvironment is different from the one of normal cells, as it promotes tumor growth, has high levels of cross-linked collagen, and presents with increased integrin signaling [133]. Because of the limited understanding of the properties of the microenvironment, effective tumor-cell killing amount in the tumor itself is not always reached; this leads to conventional drug delivery system failing. Microenvironment properties that can be used for targeting purposes with nanoparticles are hypoxia, mild acidity, angiogenesis blockage and nutrient depletion [42,133]. It is worth mentioning that die to tumor heterogeneity among humans the preclinical nanotherapeutic studies of targeting tumor microenvironment will be larger than the ones that will move to the clinic [133].

## 5. Nanomedicine

Nanomedicine is still a developing field which is based on the use of physical properties of nano-sized molecules or nanoparticles in health and medicine for disease prevention, diagnostic purposes and therapeutic applications at the molecular level [5,27]. Nanomedicine combines nanotechnology with pharmacology and biomedicine with the aim of developing drugs and imaging agents with high efficacy, improved safety and toxicology profiles [12,123]. Nanomedicine can be divided into five subfields: analytical tools, nanoimaging, nanomaterials and nanodevices, novel therapeutics and drug delivery systems, and clinical, regulatory and toxicological issues [134]. Nanomedicine relies on the development of nanorobots and molecular machines that are able to identify pathogens and replace cells or cellular components in vivo [135]. The ability of nanomedicine to integrate the advances of genomics and proteomics into molecular medicine will lead to the development of personalized medicine. This will allow for better understanding of diseases at the molecular level, which will lead to patient-tailored therapies (Figure 3). Personalized medicine is the prescription of therapeutics based on patients’ genetic factors and outer influences [136]. Personalized medicine will consider pharmacogenetic, pharmacogenomic, pharmacometabolic and pharmacoproteomic patient information combined with environmental factors that can influence response to therapy [6,129,131]. Molecular biomarkers which are defined as alteration at DNA, RNA, metabolite and/or protein levels, are important in personalized medicine, for both diagnostics and therapy [136]. Biomarkers are characteristics that can be objectively measured as indicators of physiological and pathological processes, and can serve as indicators for pharmacological response to a treatment method [6]. As mentioned before, nanoparticles are suitable for biolabeling, which is crucial for biomarker discovery [137]. In the future, development of multifunctional and multipurpose nanoparticles will accelerate the progress of precision medicine.

Nanoparticles shed new light in the medical field as physiological and pathological cellular processes occur at the nanoscale [5]. Nanoparticles specificity can be influenced with different internal patho-physiological or patho-chemical conditions (e.g., pH, ionic strength, hypoxic environment) and external (e.g., temperature, light, ultrasound) stimuli [10,11]. Nanoparticles are attractive for the research community as they provide higher sensitivity at lower costs [126]. Incorporation of drug properties to nanoparticles (i.e., nanomedicines) can modify the effect of the drug and change its biodistribution and dosing efficacy in the targeted site with the purpose of obtaining superior therapeutic outcome and reduced side effects [9,27]. The majority of the nanomedicines are administered intravenously, although there are preclinical studies for developing nanomedicines for oral administration and brain drug delivery through the oral route [27,138,139]. Although the majority of nanoparticles are developed for targeting human malignancies, several are explored for addressing the clinical challenge of inflammatory diseases like rheumatoid arthritis, inflammatory bowel disease, asthma, multiple sclerosis, diabetes and neurodegenerative disorders [13]. 

### 5.1. Nanooncology

Conventional cancer therapy consists of surgical removal of the tumor which is followed by radiation and chemotherapy. However, due to low specificity, limited targeting, rapid drug clearance and biodegradation, the therapy does not always have positive results [10]. However, the favorable properties of nanoparticles like small size, great surface-to-volume ratio and monitored drug release implicate their potential in better and more targeted treatment of oncological patients [10,11]. Nanoparticles show potential to modulate pharmacokinetics and pharmacodynamics properties of drugs, thus enhancing their therapeutic potential [11].

The application of nanotechnology in the field of oncology is termed “nanooncology” and includes both diagnostics and therapy [6]. Therefore, different lipid-based, polymer-based, inorganic, viral and drug-conjugated nanoparticles are explored for potential use in oncology. So far, gold nano-shells, iron oxide nano-crystals and quantum dots based nanomedicines have been developed for use in oncology [9]. In the nanooncology field, drug delivery can either be passive or active. Passive relies on the increased vascular permeability and retention effect to achieve drug accumulation on the tumor cells, while active refers to the accumulation of targeted nanoparticles at the site of interest with specific active drug delivery strategies through different molecular recognition forms such as lectin-carbohydrate, ligand-receptor or antigen-antibody [9,123,140,141]. Molecular recognition will allow the controlled release of an encapsulated drug only after the nanoparticle has reached its target molecule. This will result in higher specificity and lower toxicity. To be used in clinical care, nanoparticles should be biocompatible, biodegradable and nontoxic, stable after administration, and easily produced at a large scale with control over their physico-chemical properties [13]. For treatment of solid tumors, usually passive targeting is used because of the increased permeability of blood vessels and poor lymphatic drainage that allow for drug accumulation within the tumor microenvironment due to the EPR effect [123,132]. It is reasoned that, for successful therapy, nanoparticles should be between 5 and 200 nm in size, as those smaller than 5 nm will be rapidly eliminated by urinary excretion, and those larger than 200 nm can be captured by the reticuloendothelial system of the liver and spleen [42]. For passive targeting, drug delivery systems should be less than 100 nm in diameter with hydrophilic surface [141]. Nanotechnology-based products for cancer treatment are already approved by the FDA [12]. Such examples are a liposome preparation of doxorubicin named Doxil, Caelyx or Myocet, liposomal daunorubicin—DaunoXome, liposomal daunorubicin-cytarabine combination—Vyxeos, liposomal cytarabine—DepoCyt, liposomal encapsulated cisplatin—Lipoplatin, liposomal irinotecan—Onivyde, and a nanoparticle formulation of paclitaxel named Abraxene—all of which are forms of passive targeted first generation nanomedicines [5]. Various oncology nanomedicines (eg. thermally sensitive liposome with encapsulated doxorubicin—ThermoDox) are still in clinical trials [9]. In cancers in general, three major sets of cellular targets for active targeting are considered: cells which overexpress receptors for transferrin, folate, epidermal growth factor or glycoproteins; tumor endothelium overexpressing vascular endothelial growth factor, integrins, vascular cell adhesion molecule-1, or matrix metalloproteinasas; and cells stromal cells that can acquire tumor survival-propagating phenotype [13]. Despite the growing number of nanomedicines in research studies, only a small set of them reach clinical application. This is partially influenced by tumor heterogeneity where “one size fits all” mechanisms are difficult to be applied [13]. Other reasons are limited specificity, accumulation and specificity, and the use of inappropriate animal models, which lead to unsuccessful translation into humans. To be granted an approval, nanomedicines that are based on existing FDA-approved drugs must show improvements like lower toxicity and greater efficacy over the existing drug. 

### 5.2. Challenges, Social Concerns and Safety Issues of Nanomedicine

Liposomal preparations which represent the first generation of nanomedicines are used without toxicity so far. However, the nature of other nanomedicines like phospholipids or biodegradable polymers may introduce toxic effects of which we are yet unaware [5]. The origin of the nanomaterial (naturally occurring or manufactured) will likely be an important variable in the determination of possible toxic effects. Although there is no conclusive evidence of toxicity in humans so far, it is anticipated that carbon nanotubes can induce asbestos-like inflammation, while the heavy-metal composition of nanomaterials and their ability to enter and possibly accumulate in vital organs (liver, brain, lungs) can lead to tissue-specific toxicity [123,127]. Although a single dose of nanoparticle administration is below the limit for heavy-metal poisoning, our inability to break them down can present consequences we cannot assess [128]. For example, higher doses or long exposure to gold nanoparticles are toxic for humans as they accumulate in the blood and tissues due to a low clearance rate. Gold nanoparticles pose additional risks due to their affinity for DNA [123]. Metallic nanoparticles can be considered toxic as they generate reactive oxygen species; in addition, they remain in the environment for long time-periods, which leads to constant human exposure with unknown consequences. The toxic effects will vary depending on nanomedicine size, shape, structure, biocompatibility, degradation, chemical composition, solubility and aggregation. Another factor to be considered is their administration; depending on the type of administration (dermal, oral, respiratory or intravenous), different risks are to be considered. For example, for systemic administration (intravenous injection), the compatibility of nanomedicines with blood cells, blood coagulation and platelet function should be tested.

Although the implementation of nanoparticles in medicine is with the aim to increase the benefits of the patients from available therapy, their use raises questions about the risks they impose to human subjects and the environment [127]. Once administered, it is still unclear whether the properties of nanomaterials will change when in contact with specific microenvironments, how will they affect the oxidative and inflammatory responses, how will they influence complement and phagocyte activation, coagulation effects as well as human immunity [123,127]. It is also uncertain if their main function will be altered once they come in contact with biological fluids or molecules. As the physicochemical properties of nanomaterials are not fully understood, their involvement in clinical trials should be carefully monitored. For safety reasons, nanoparticles should be characterized with multiple methods. This encourages the development of a new specialized field of study named *nanotoxicology* which should adequately evaluate potential nanoparticle toxicity with more appropriate and sensitive methods [11]. For determining the properties, biological activity and toxicity of nanoparticles’ new nano-compatible assays will have to be developed [123]. Before starting a clinical trial, researchers can minimize the risks by conducting an extensive literature review on their subject of interest to understand potential side-effects (if published), strictly defining inclusion and exclusion criteria for individuals likely to be affected by the study, closely monitoring the results during and after the trials and reporting to the authorities and review boards [127]. To reduce potential damage, dose ranges should be determined, pharmacokinetics of nanomaterials should be examined, and their absorption, distribution, metabolism and excretion should also be studied using in vitro and in vivo experiments. Concerns can be raised due to the anatomical and physiological differences between animals and humans—i.e., how laboratory animals (rodents) and human subjects react to and metabolize substances; what is non-toxic to animals can be toxic in humans and vice versa. Another limitation is the time required to determine if a substance is toxic or not. In humans, it may take decades for the toxic effects to be detected. Ideally, for accurately determining toxicity, human subjects should be exposed to the desired amounts of nanomaterials for a relatively long time period. Frequent administration and continuous need for structurally complex nanoparticles for such studies requires larger amounts of purified medicines. This may present as a problem as batch-to-batch nanomedicines should be tightly controlled and regularly tested for their physico-chemical properties, which makes the industrial process more demanding and expensive [11]. Nanomedicines should prove enough stability, long shelf life and stability during long-term storage and clinical administration [13]. Additionally, human biological diversity, disease heterogeneity, existence of biological barriers and accumulation in non-target organs should be taken into consideration [13]. More importantly, research subjects involved in clinical trials should be aware of the possible benefits and risks they undertake when participating in such studies. Moreover, possible effects on second (family, fetuses, breast-fed infants) and third (research team, staff) parties should be taken into consideration as well. Finally, before starting clinical trials, investigators should compare possible risks with benefits—if the expected benefits for the studied subject or society are greater than the risks, in general, the study should be justified.

Another issue which can arise is the intellectual property. As nanomedicines are complex structures consisting of several components, it should be determined who the intellectual property rights belong to [13]. The intellectual property right for the carrier, the technology, and the encapsulated cargo (drug) and carrier together should be clearly defined. This can be further complicated with the use of already existing drugs in combination with new technologies. As nanomedicines are the bridge between nanotechnology and medicine, legal regulations should make it easier for these two branches to cooperate. The formation of hybrid scientific fields will greatly improve the development of nanotechnology and nanomedicine fields as specialists from different disciplines and backgrounds will cooperate in order to solve the problems arising. Ultimately, government policies should be implemented for the regulation of nanomedicines at the global market. As they have the potential to greatly influence the biopharmaceutical market their standardization (safety, efficacy, nomenclature, quality) and commercialization should be tightly regulated. These regulations ought to take into consideration all biological variables that can be encountered when using nanomedicines in clinical trials and further on in clinical practice.

## 6. Conclusions

Nanotechnology has seen significant advancement in the past several decades. It is believed that it has a considerable capacity for development of new and original approaches in clinical oncology. As the 5-year survival rate of cancer patients still remains poor, the need for new cancer treatment concepts is highly needed. From the perspective of personalized medicine, nanotechnology has great potential for diagnosis, therapy and treatment. It offers the possibility to tailor the treatment regimens for each patient separately, and it gives unique options for managing treatment of inaccessible tumors and for targeting and elimination of residual tumor mass and/or cells which are the main players in the recurrence of cancer.

In the future, nanomaterials are expected to surpass current cancer therapy consisting of operative removal of tumors, chemotherapy and radiation therapy. Nanocarriers have shown many advantages in comparison to free drugs, since they can protect drugs from early degradation and premature interaction with biological environments while reaching target cells and/or tissues. With the use of nanocarriers, the pharmacokinetic and drug distribution profile can be better regulated, and nevertheless the absorption and intracellular penetration of the drugs in target cells/tissue can be improved [3]. However, when considering a nanocarrier for use in medicine, it has to be well characterized, biocompatible, and biodegradable, be water soluble or form colloids in aqueous conditions, have extended circulating half-life and also long shelf life. Nevertheless, it has to target specific cells and/or tissues [3]. The potential toxicity issues of the actual application of nanomaterials in patient clinical care are not negligible and should be considered carefully. It would be preferential for nanomaterials to be additionally engineered to be biocompatible and biodegradable. In addition, accumulation in human off-target organs should be avoided and clearance rates should be improved in order for minimal damage to be caused upon application. Nanomaterials offer a wide range of different combinations in production of nanoparticles; however, we have to keep in mind that in the end they have to be batch to batch comparable, simple to assemble and have to withstand rigorous clinical tests in order to prove their specificity and safety. Finally, the footprint on the environment resulting from the broad use of nanomaterials should be carefully determined and thoroughly assessed.

## Figures and Tables

**Figure 1 materials-12-01588-f001:**
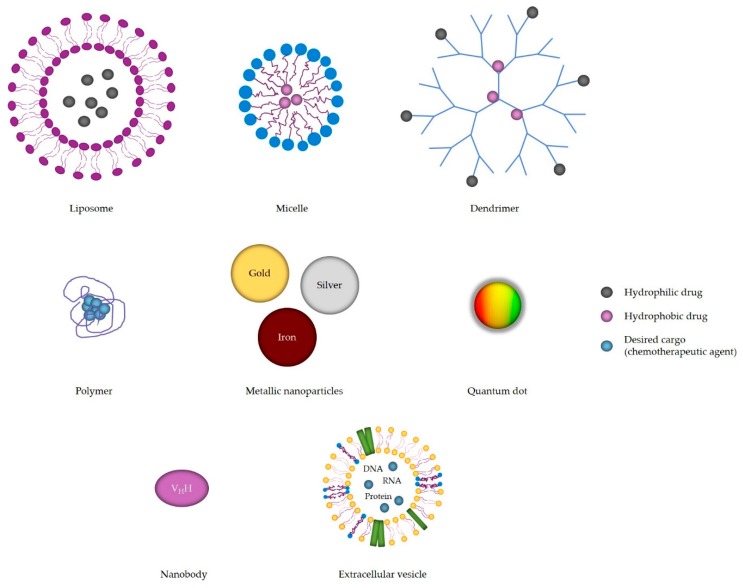
Schematic representation of the structures of different nanoparticles: organic (liposomes, micelles, dendrimers and polymers), metallic (gold, silver and iron nanoparticles, and quantum dots) and biological (nanobodies and extracellular vesicles–exosomes). The image is for graphical illustration only and does not represent actual sizes or size ratios among particles.

**Figure 2 materials-12-01588-f002:**
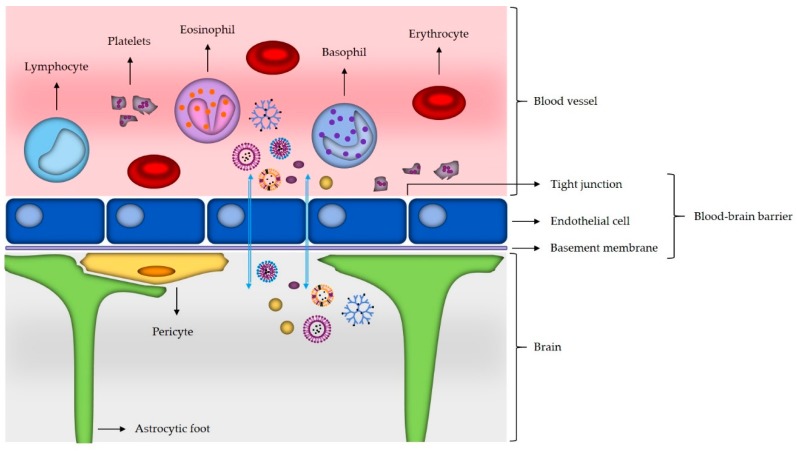
Schematic representation of the blood–brain barrier (BBB). Nanomaterials that are able to cross the BBB (gold nanoparticles, liposomes, micelles, dendrimers, exosomes and nanobodies) are illustrated. Blue arrows show transport through the BBB. The image is for graphical illustration only and does not represent actual sizes or size ratios among particles.

**Figure 3 materials-12-01588-f003:**
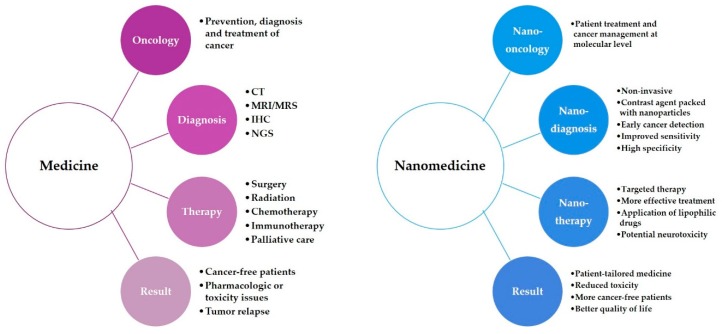
Medicine versus nanomedicine. The scheme draws a parallel between classical medicine and the evolving nanomedicine. The major ways for disease diagnosis and treatment are outlined for both fields. Patient benefit from both approaches is also indicated. In contrast to classical medical care which in glioblastoma especially leads to frequent recurrence and short life expectancy, development of patient-tailored treatment in nanomedicine will lead to more cancer-free patients with prolonged life expectancy i.e., better quality of life. CT—computer tomography; MRI—magnetic resonance imaging; MRS—magnetic resonance spectroscopy; IHC—immunohistochemistry; NGS—next generation sequencing.

**Table 1 materials-12-01588-t001:** Summary of the advantages of nanomaterials.

Nanomaterial	Advantages
Silver nanoparticles	Bactericidal properties Antiviral function
Gold nanoparticles	Plasmon resonance Absorb light in near infrared region Easy surface-conjugation with antibodies Suitable for passive and active targeting Can be used as drug carriers CT contrast agent
Magnetic nanoparticles	Active when external magnetic field is applied Selective destruction of cancer cells in hypoxic areas as a result of heat release
Platinum nanoparticles	Protection mechanisms against ROS
Quantum dots	Bright Photostable Broad excitation and narrow emission spectra Signal amplification is not needed
Liposomes	Suitable for packing neutral, hydrophilic and hydrophobic drugs Engineered to release cargo upon suitable pH, redox potential, ultrasound and electromagnetic field Passive and active targeting PEG-coating increases biocompatibility, water solubility and half-life, and lowers toxicity Suitable as imaging tools (CT)
Block copolymere micelles	Carriers of water-insoluble drugs Stable High loading capacity
Dendrimers	Monodispersity Very small size PEG-conjugation decreases toxicity, enhances biocompatibility and EPR effect, and increases half-life Slow drug release Controlled release upon pH, glutathione or enzyme stimuli High tumor accumulation Suitable for antibody and nucleic acid delivery Use in diagnostics as contrast agents in MR, CT and fluorescence imaging
Polymers	Improved pharmacokinetic and pharmacodynamic characteristics Controlled drug release upon diffusion-control, solvent-activation, chemical control or external triggers (pH, temperature and redox potential) Biodegradable, biocompatible and non-toxic
Nanobodies	High antigen affinity and specificity Economic production High stability at elevated temperatures and non-physiological pH Water solubility Low immunogenicity PEG- or albumin-binding increases half-life Better tumor penetration and distribution Suitable for use in PET and SPECT
Extracellular vesicles	Carriers of different cell proteins, viral proteins, nucleic acids and lipids Biocompatible Drug carriers Less toxic and immunogenic Present in blood and cerebrospinal fluid

PEG—polyethylene glycol, CT—computer tomography, ROS—reactive oxygen species, EPR—enhanced permeability and retention, MR—magnetic resonance, PET—positron emission tomography, SPECT—single photon emission computed tomography.

**Table 2 materials-12-01588-t002:** A list of glioblastoma clinical trials based on the use of nanomaterials.

Disease	Agent	Clinical Trial Number	Phase
Recurrent high grade glioma Newly diagnosed glioblastoma	ABI-009 (Nab-Rapamycin)	NCT03463265	II
Recurrent high grade glioma	NL CPT-11 (Nanoliposomal CPT-11)	NCT00734682	I completed
Recurrent high grade glioma	Ferumoxytol	NCT00769093	I
Glioblastoma	9-ING-41	NCT03678883	I/II
Recurrent high grade glioma	Liposomal irinotecan	NCT02022644	I
Recurrent malignant glioma or solid tumors and brain metastases	2B3-101	NCT01386580	I/II
Children and adolescents with refractory or relapsed malignant glioma	Myocet	NCT02861222	I
Glioblastoma and diffuse intristic pontine glioma	Doxorubicin	NCT02758366	II
Recurrent glioblastoma or gliosarcoma	NU-0129	NCT03020017	I
Recurrent glioblastoma	SGT-53	NCT02340156	II
Recurrent glioblastoma	RNL (rhenium nanoliposomes)	NCT01906385	I

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
