# Peer review of "Nanotechnology Meets Oncology: Nanomaterials in Brain Cancer Research, Diagnosis and Therapy"

_materials, 2019, doi:10.3390/ma12101588_

Reviewer 1 Report

The authors review the recent advances in nanomaterials based brain cancer research. It’s an important and rapidly developing research field, I think this timely review could be greatly beneficial to the related research fields. The manuscript was well-written, I only have a few suggestions here.

1, Since this review mainly focuses on brain cancer diagnostic and therapy, perhaps the authors could move the introduction of glioblastoma to the beginning of the paper. It’s a matter of test but I think it flows better if to start with the general introduction of Gli and then summarize the major challenges in the brain cancer therapy, and then followed by the entrance of nanotechnology-based therapies.

2, Some important references are missing when discussing present therapy options for glioblastoma, e.g. the newly emerged immunotherapy and antibody–drug conjugates based therapies (Nat. Rev. Clin. Oncol., 2018, 15, 422).

3, Some recent literature describing nanotechnology-based brain cancer therapy are missing (with innovative new concepts and capabilities), please consider citing: Adv. Drug Deliv. Rev., 2019, 138, 344;  Adv. Mat., 2018, 30, 1705054.

4, The toxicity of nanoparticles is one of the most important aspects considering their actual application and clinical translation, it would be great if the authors could add some more comments in the conclusions/outlook section.

Author Response

Comment: Since this review mainly focuses on brain cancer diagnostics and therapy, perhaps the authors could move the introduction of glioblastoma to the beginning of the paper. It is a matter of taste but I think it flows better if you start with the general introduction of glioblastoma and then summarize the major challenges in the brain cancer therapy, and then followed by the entrance of nanotechnology-based therapies.

Response: As suggested by the reviewer, we changed the order of the sections. The manuscript now starts with the general introduction of cancers “Cancer: from macro to nano” and is followed by the section “Glioblastoma” which briefly outlines the disease and its current problems.

Comment: Some important references are missing when discussing present therapy options for glioblastoma, e.g. the newly emerged immunotherapy and antibody-drug conjugates based therapies (Nat. Rev. Clin. Oncol., 2018, 15, 422).

Response: We have added the suggested reference in the section “Glioblastoma”.

Comment: Some recent literature describing nanotechnology-based brain cancer therapy are missing (with innovative concepts and capabilities), please consider citing: Adv. Drug Deliv. Rev., 2019, 138, 344; Adv. Mat., 2018, 30, 1705054.

Response: We have added the suggested references in the section “Glioblastoma”.

Comment: The toxicity of nanoparticles is one of the most important aspects considering their actual application and clinical translation, it would be great if the authors could add some more comments in the conclusions/outlook section.

Response: In the manuscript we have dedicated a whole section which discusses the possible toxicity issues of the broad application of nanomaterials for human subjects and the environment. However, we thank the reviewer for this comment and ask him to see the changes we have introduced in the Abstract and Conclusions sections.

Reviewer 2 Report

Dear authors

The paper summarized the recent advanced discoveries in the field of Nano Oncology with a special interest in brain cancer research. The review is well written and easy to follow.

Minor points

In general, the manuscript is written without going in details of the described approaches. To improve the interest of the manuscript, I suggest to add more details of the applications of nanomaterials in brain cancer. You can discuss why different nanomaterials can cross the BBB. A table which summarize the advantages of each nanomaterials should be added.

Natural vesicles such as exosome could be included (e.g. PMID:21423189, PMID:27558906)

References are updated.

Author Response

Comment: In general, the manuscript is written without going in details of the described approaches. To improve the interest of the manuscript, I suggest to add more details of the applications of nanomaterials in brain cancer. You can discuss why different nanomaterials can cross the BBB.

Response: Please see the new section “Mechanisms of nanomaterials transport across the blood-brain barrier”.

Comment: A table which summarizes the advantages of each nanomaterial should be added.

Response: Please see Table 1.

Comment: Natural vesicles such as exosomes should be included (e.g. PMID: 21423189, PMID: 27558906).

Response: Please see the new section “Extracellular vesicles”. We have also updated Figure 1 and the section “The use of nanoparticles in glioblastoma targeting” and included information about extracellular vesicles.